# Additive Manufacturing of a Special-Shaped Energetic Grain and Its Performance

**DOI:** 10.3390/mi12121509

**Published:** 2021-12-04

**Authors:** Yongjin Chen, Shuhong Ba, Hui Ren

**Affiliations:** 1State Key Laboratory of Explosion Science and Technology, Beijing Institute of Technology, Beijing 100081, China; 2School of Equipment Engineering, Shenyang Ligong University, Shenyang 110159, China; shuhongba@163.com

**Keywords:** energetic material, additive manufacturing, 3D printing technology, aluminum and ammonium perchlorate, special-shaped energetic grain, combustion performance

## Abstract

In order to solve the problems of the complicated forming process, poor adaptability, low safety, and high cost of special-shaped energetic grains, light-curing 3D printing technology was applied to the forming field of energetic grains, and the feasibility of 3D printing (additive manufacturing) complex special-shaped energetic grains was explored. A photocurable resin was developed. A demonstration formula of a 3D printing energetic slurry composed of 41 wt% ultra-fine ammonium perchlorate (AP), 11 wt% modified aluminum (Al), and 48 wt% photocurable resin was fabricated. The special-shaped energetic grains were successfully 3D printed based on light-curing 3D printing technology. The optimal printing parameters were obtained. The microstructure, density, thermal decomposition, combustion performance, and mechanical properties of the printed grain were characterized. The microstructure of the grain shows that the surface of the grain is smooth, the internal structure is dense, and there are no defects. The average density is 1.606 g·cm^−3^, and the grain has good uniformity and stability. The thermal decomposition of the grain shows that it can be divided into three stages: endothermic, exothermic, and secondary exothermic, and the Al of the grain has a significant catalytic effect on the thermal decomposition of AP. The combustion performance of the grain shows that a uniform flame with a one-way jet is produced, and the average burning rate is 5.11 mm·s^−^^1^. The peak pressure of the sample is 45.917 KPa, and the pressurization rate is 94.874 KPa·s^−^^1^. The analysis of the mechanical properties shows that the compressive strength is 9.83 MPa and the tensile strength is 8.78 MPa.

## 1. Introduction

Special-shaped energetic grains can realize special functions and needs due to their special shape [1], complex structure, and partial or full height symmetry, which makes their demand in the fields of weaponry and aerospace more urgent [2]. For example, in the field of weaponry, propellant is used as the energy source for barrel weapons to launch projectiles, and its composition and structure are the key factors that determine the power of the barrel weapon [3]. Improving ballistic efficiency is an important means to improve the power of a weapon [4]; at present, the main way to improve the ballistic efficiency of a propellant is to control the burning rate and burning surface of the propellant, and this requires special-shaped propellant grains to achieve. Similarly, in the field of aerospace, propellant grains are used as the power source for propulsion systems, and different shaped propellant grains need to be designed to achieve different working capabilities, thereby achieving the controllable flight attitude and adjustable flight speed of the aircraft [5,6,7].

For the forming of special-shaped energetic grains, there is mainly the melting-casting method, the casting method, and the press-stretching method [8]. Among them, the grains obtained by the melting and casting method have the disadvantages of a long solidification time and limited life of the mold tank, and there may be some defects such as pores or cracks in the grain during casting. In addition, for the forming of complex special-shaped grains, corresponding molds need to be made, which greatly increases the production and manufacturing costs. At the same time, the requirements for processing technology and mold manufacturing are relatively strict, and it is difficult to meet the ideal requirements. Extrusion molding may increase the local pressure of grains, which is dangerous to some extent. At the same time, human-machine isolation cannot be completely realized. It can be seen that the forming and manufacturing process of special-shaped energetic grains generally has defects such as complicated and tedious processes, the poor adaptability of special-shaped grains, poor process safety, etc. Therefore, it is necessary to improve the manufacturing technology of energetic materials. 3D printing technology is a kind of technology based on the 3D model data of the sample which uses a high-energy beam source or other methods to stack and bond liquid, powder and other materials layer by layer, creating the final form by stacking [9]. 3D printing technology has been successfully applied in various fields [10,11], including military [12,13] and aerospace fields [14,15,16]. At present, the additive manufacturing technologies applied in the field of energetic material forming mainly include direct writing technology [17,18,19,20,21,22,23], melt extrusion technology [24], inkjet deposition printing technology [25,26,27], screen printing technology [28], and electrophoretic deposition technology [29]. The energetic materials obtained by these technologies are mostly planar structures or microstructures [30,31,32,33,34,35,36,37]. Although there are very few printed three-dimensional structures [38,39,40], the dimensional tolerance and structural accuracy of shaped grains have a certain gap between the ideal model of the design [41,42]. As an additive manufacturing technology, light-curing 3D printing technology can make up for the shortcomings of the previous technologies, and it has the characteristics of fast printing speed, high precision, the ability to free-form complex structures, green production and so on [43,44]. Therefore, light-curing 3D printing technology is applied to the forming field of energetic materials to realize an efficient, safe, green, convenient and automated mold-free production pattern of complex special-shaped energetic grains.

There is little research on the application of light-curing 3D printing technology in the forming field of energetic materials. The Netherlands Organisation for Applied Scientific Research (TNO) has done some work on the manufacture of “LOVA propellant” by using stereolithography (SLA) [45,46]. TNO researchers used SLA to print a photocurable energetic resin composed of 50 wt% RDX and 50 wt% UV polymerizable acrylate binder, and quincunx propellant grains with 19 longitudinal and radial perforations were obtained. Then, the binder system was improved. The energetic resin system with the formula of 50 wt% RDX, 25 wt% acrylate binder, and 25 wt% energetic plasticizer was printed, and a new type of high bulk density propellant grain sample with 14 perforations was obtained. The combustion performance of the 3D-printed ribbon propellant was tested. At the same time, a stack of porous disk propellants with a diameter of 29 mm with radial holes were obtained, and the ballistic test was carried out on a Gau-8 (Goalkeeper) gun system with a 30 mm cartridge chamber with a muzzle velocity of 26–370 m/s. However, due to the use of a large number of non-energetic binders, the power of gunpowder propellant is low; it is only at the level of a single-base propellant [47]. A 3D-printed gun propellant based on stereolithography (SLA) was developed by Weitao Yang and Rui Hu et al. [48,49], and a demonstration formula composed of 50 wt% 1,3,5-Trinitro-1,3,5-triazinane (RDX), 25 wt% epoxy acrylate, 12.5 wt% N-n-butyl-N-(2-nitroxyethyl) nitramine (Bu-NENA), 12.5 wt% reactive diluent and additives was fabricated. Porous propellants with a thickness of 3 mm and outer diameter of 40 mm were obtained. The mechanical properties and combustion performance of the propellant were tested, and the gun launch experiment was carried out. McClain M.S et al. combined direct writing technology with light-curing technology [50], used hydroxyl-terminated polybutadiene (HTPB) and a UV-curable polyurethane binder, printed ammonium perchlorate (AP) composite propellants at 85% solids loading, and obtained a rectangular grain.

In this paper, a photocurable resin was developed. A demonstration formula of 3D-printed energetic slurry composed of 41wt% ultra-fine ammonium perchlorate (AP), 11 wt% modified aluminum (Al), and 48 wt% photocurable resin was fabricated. The special-shaped energetic grains were successfully 3D printed based on light-curing 3D printing technology. The optimal printing parameters were obtained. The microstructure, density and uniformity, thermal decomposition, combustion performance and mechanical properties of the printed grains were characterized.

## 2. Experimental Section

### 2.1. Experimental Materials

Aluminum powder (Al) (AR grade) with a particle size of 1–5 µm was purchased from Shanghai Chaowei Nano Technology Co., Ltd., Shanghai, China. Ammonium perchlorate (AP) (AR grade) with a particle size of 5–8 µm was purchased from Dalian North Potassium Chlorate., Co., Ltd., Dalian, China. Epoxy acrylic (EA) and polyurethane-acrylic (PUA) were purchased from DSM (China) Co., Ltd., Beijing, China. 1,6-hexanediol diacrylate (HDDA) was purchased from Shanghai Guangyi Chemical Co., Ltd., Shanghai, China. Trimethylolpropane triacrylate (TMPTA) was purchased from Taiwan Changxing Chemical Co., Ltd., Taiwan, China. Diphenyl (2,4,6-trimethylbenzoyl)phosphine oxide (TPO) was purchased from BASF (China) Co., Ltd., Shanghai, China.

### 2.2. Characterization and Testing

The grains were prepared by a light-curing 3D printer (Self-built). Density was measured by densitometer (AU-120S, Hangzhou Goldmic Instrument Co., Ltd., Hangzhou, China). The microstructure of the grain was observed by a field emission scanning electron microscope (FE-SEM, Ultra-55, Carl Zeiss, Oberkochen, Germany) and video microscope (PMS-XHD-AF, Pulmis Precision Instrument Co., Ltd., Dongguan, China). The EDS energy spectrum was measured by an x-ray energy spectrometer (EDS, INCA 100 Hitachi, Tokyo, Japan). Mechanical properties were measured by the electronic universal testing machine (UTM4304, Shenzhen Sansi Zongheng Technology Co., Ltd., Shenzhen, China). The DTA curve was measured by a high-temperature differential thermal analyzer (DTA, CRY-2P, Shanghai jingke tianmei scientific instrument Co., Ltd., Shanghai, China). The burning flame shape was recorded by a high-speed camera (UX100, Photron, Tokyo, Japan). Burning velocity was calculated after recording the time with a Tektronix oscilloscope (TBS1102B, Shenzhen Zhongru Electronics Co., Ltd., Shenzhen, China).

### 2.3. Formula and Printing Process Parameters

#### 2.3.1. Formula and Preparation of Energetic Slurry

Using self-made photocurable resin (formula as shown in Table 1) as the binder, modified aluminum powder Al (TEM, Figure 1a) and ultra-fine AP (TEM, Figure 1b) as solid filler, the energetic slurry was prepared according to the technological process of Figure 2. According to the basic formula of HTPB propellant (HTPB/AP/Al = 14 wt%/68 wt%/18 wt%) [51], the formula of energetic slurry is obtained; that is, photocurable resin/AP/Al = 48 wt%/41 wt%/11 wt%. The formula of the energetic slurry is shown in Table 2.

#### 2.3.2. Optimization of Printing Parameters

Energetic slurries with solid filler contents of 40%, 50% and 60% was prepared, and the slurry with different solid contents was printed. The shaped grains were shown in Figure 3a–c. According to the precision and combustion situation of shaped grains, the relationship between the slurry solid content and the printing process parameters was explored.

It can be seen from Figure 3 that with the increase of solid content, the precision of the shaped grain gradually weakens. When the solid content is 40%, the grain sample has high fidelity and high forming precision, but due to the high resin content, the combustion performance is poor so, the grain is not easily ignited. When the solid content is 50%, the grain basically restores the shape of the model, which can be ignited and meet the basic needs. When the solid content increases to 60%, the grain sample is deformed, the layers fall off, and the grain precision is poor.

At the same time, the printing parameters were adjusted, and it was found that when the solid content of slurry was 52%, the precision and combustion performance of the printed grain were relatively good. The specific parameters of 3D printing are shown in Table 3. In order to ensure that the shaped grain was firmly bonded to the printing platform, the first 10 layers (each layer with a slice thickness of 100 µm) need a longer exposure time. In the later stage of printing, each layer was exposed for about 60 s, and the slurry can be completely cured. According to the data in Table 3, it takes about 1.9 h to print a grain with a height of 3 cm.

### 2.4. Experimental Process of Light-Curing 3D Printing

The light-curing 3D printing technology uses a digital light processor to irradiate and cure the photopolymer while curing the entire cross-section. It is in the way of the surface layer by layer superposition until the specimen is finally printed and formed. A schematic diagram of light-curing 3D printing technology is as shown in Figure 4.

The light-curing 3D printing forming process of special-shaped energetic grains mainly includes four processes: construction of the three-dimensional model, pre-processing, the printing and curing process, and post-processing, as shown in Figure 5.

According to the experimental steps in Figure 5 and the designed 3D models of special-shaped grains, the energetic slurry with a solid content of 52% was printed. The specific printing process is shown in Figure 6a, and the printed energetic grains are hung on the printing platform (red area). After printing, the formed samples are post-processed to obtain different special-shaped grains, as shown in Figure 6b.

The printed real samples of different special-shaped grains were compared with the three-dimensional CAD models of special-shaped grains, as shown in Figure 7 (the light color represents grain 3D models, while the dark color represents real grain samples).

It can be seen from Figure 7 that the real samples of energetic grain printed by the light-curing 3D printing technology highly and truly restored the characteristics of the 3D models of the grains with high printing accuracy and fast forming speed.

## 3. Results and Discussion

### 3.1. Density and Uniformity Test

According to the principle of drainage method, the density of energetic grain was tested by a densitometer. The density uniformity was considered by testing the density of the upper, middle and lower parts of the grain, and the standard deviation was calculated. The density was measured according to the GJB772A-97. The test results are shown in Table 4.

It can be seen from Table 4 that the average measured density of the grains is 1.606 g/cm^3^, and according to the formula, the theoretical density of the grains is calculated to be 1.619 g/cm^3^, which has little difference when compared with the measured density. The standard deviation of the experimental density of the 3D-printed grains is 0.002 g/cm^3^, which indicates that the density of the grain has better uniformity.

### 3.2. Surface and Internal Structure

The surface structure of the 3D-printed grain (Red block diagram area in Figure 8a) was observed by optical microscopy and scanning electron microscopy (SEM). The results are shown in Figure 8b,c respectively.

It can be seen from Figure 8 that the surface of the 3D-printed grains is flat, smooth, and without obvious defects. In addition, according to the SEM of a grain surface in Figure 8c, when the grain surface is further enlarged to 10,000 times, the grain surface is still smooth, and there are no obvious cracks and other defects, which indicates that the 3D printed grain surface has high flatness and integrity.

Figure 9 shows the SEM of the internal structure of the 3D-printed grain. Figure 9a,b shows the microscopic morphology of the cross-section of the grain under different magnifications. Figure 9c,d shows a further detail of the cross-sectional morphology of the corresponding position (red circle). In order to further characterize the distribution of elements in the grain, EDS analysis was carried out on the purple block diagram in Figure 9b, as shown in Figure 9e, and the element distribution results are shown in Figure 9 (e1–e5).

It can be seen from Figure 9a,b that the 3D-printed grains have good cross-section integrity, compact structure, and no obvious cracks, holes and other defects. In addition, it can be seen from Figure 9c,d that no obvious cracks, holes and other defects are still found in the grain cross-section under a larger magnification, and the cross-section is relatively complete as a whole. It can be seen from Figure 9(e1–e5) that Al, Cl, C, N and O elements are uniformly distributed inside the 3D-printed grain, indicating that the distribution of oxidant (AP), reductant (Al) and resin systems in the system is relatively uniform. The SEM/EDS of the cross-section of the grain shows that the light-curing 3D printing technology has achieved a compact internal structure of the energetic grain and reduced the formation of defects; at the same time, all elements in the grain were uniformly distributed.

### 3.3. Mechanical Property Test

Using an electronic universal testing machine, the compressive and tensile properties of the printed samples were tested at 25 °C and a relative humidity of 50%. The compressive strength of a cylindrical grain with a size of 20 mm × 30 mm was tested with a loading rate of 2 mm/min. The maximum load Q_c_ on the compression load-displacement curve of each test represents the compressive load of the grain, and the effective compression distance ΔL_c_ of the grain is expressed by the distance between the displacement at 10% Q_c_ and the displacement at Q_c_. According to Q_c_ and ΔL_c_, the compressive strength S_c_ and compression ratio of each test were obtained. The measurement was carried out according to the standard GB/T 2569-1995. The compressive load-displacement curve of the grain is shown in Figure 10. A dumbbell-shaped specimen with a size of 200 mm × 20 mm × 8 mm was tested for tensile strength, and the tensile speed was 1mm/min; the measurement was carried out according to the standard GB/T 2568-1995. The stress-strain curve of the specimen is shown in Figure 11. The relevant data of the quasi-static mechanical properties are shown in Table 5. Tensile and compressive strength tests were tested five times for each specimen, and the average value of three test data with good parallelism is selected.

It can be seen from Figure 10 that the compressive load of the 3D-printed grain increases slowly with the increase in displacement, sharply increases to its peak value, and then decreases sharply, which indicates that the grain has been deformed and broken at this time. The Q_c_ is 3089.73 N, ΔL_c_ is 0.63 mm, the compression ratio is 2.1%, and the average compressive strength is 9.83 MPa (the compressive strength is calculated according to the ratio of the Q_c_ to the initial cross-sectional area of the grain).

It can be seen from Figure 11 and Table 5 that during the tensile test of the specimen, it is found that with the increase of tensile force, the specimen only produces a very small elongation. When the tensile load increases to a certain extent (point D), a fracture occurs near the middle of the specimen, and the fracture surface is generally flat. The tensile strength at break of the specimen is 8.78 MPa, while the tensile break stress is 6.49 MPa.

To sum up, it can be seen that the compression and tensile properties of the grain formed by light-curing 3D printing are better.

### 3.4. Analysis of Thermal Decomposition Behavior

Using a high-temperature differential thermal analyzer, the differential thermal analysis of the 3D-printed grain was carried out. The heating rate was 10 °C/ min, while the raw material of the test grain is 6 mg. The differential thermal analysis (DTA) curve of the grain is shown in Figure 12.

It can be seen from Figure 12 that the grain occurs in a small range of endothermic-exothermic phenomena at 82.71 °C (red block diagram). It is speculated that this stage is the endothermic-decomposition-exothermic stage of the photocurable resin. (The glass transition temperature of the resin is 80 °C). At the same time, the DTA curve of the grain has one endothermic peak and two exothermic peaks. The endothermic peak at about 247.42 °C indicates the crystal form transformation process of AP, wherein AP changes from its orthorhombic crystal form to a cubic crystal form; the endothermic enthalpy of this process is 7.75 J/g. The first exothermic peak at 303.42 °C, which is the first stage of AP thermal decomposition (low-temperature decomposition), wherein AP partially decomposes and produces intermediate products (such as, NH_3_, etc.). In the second exothermic peak at 374.60 °C, which is the second stage of AP thermal decomposition (high-temperature decomposition), AP is completely decomposed into volatile products (such as O_2_, HCl, Cl_2_, N_2_, etc.), and the exothermic enthalpy of the entire exothermic process is 758.10 J/g. Compared with the DTA curve of pure AP (endothermic peak 247 °C, low-temperature decomposition 322.7 °C, high-temperature decomposition 477.2 °C) [52], it can be found that Al has basically no effect on the crystal transformation process of AP; however, because of the Al, the low-temperature decomposition temperature of AP was reduced by 19.42 °C. Similarly, the high-temperature decomposition temperature was reduced by 102.6 °C, which indicated that the Al in the grain has a significant catalytic effect on the low-temperature decomposition process of AP and, especially, on the high-temperature decomposition process of AP.

### 3.5. Analysis of Combustion Characteristics

A constant volume combustion test is used to evaluate the gas pressure output during the combustion of energetic materials. The peak pressure (P_max_) in the pressure-time curve is related to the gas production during the combustion of energetic materials. After igniting the sample to be measured in a constant volume combustion chamber, the sample to be measured rapidly generates its peak pressure at a certain pressurization rate. The pressurization rate is determined by the slope of the pressure-time rising curve (from 10% to 90% of the peak pressure) [53]. In this paper, the pressure-time curve of each sample is measured three times and averaged. The closed combustion chamber device for pressure-time curve test is shown in Figure 13. Figure 14 shows the pressure evolution process of the 0.5 g sample to be measured.

It can be seen from Figure 14 that the peak pressure of the sample is 45.917 KPa and the pressurization rate is 94.874 KPa/s.

A cylindrical grain with a size of Φ 5 mm × 8 mm is ignited, and the flame condition during combustion is recorded by a high-speed camera. The combustion effect in some periods is shown in Figure 15. Before the experiment, liquid paraffin was coated on the side of the cylindrical grain to ensure the grain burned on the end surfaces.

It can be seen from Figure 15 that the grain was successfully ignited, and the height of the flame is the same, indicating that the grain burning is uniform. The shape of the flame is roughly divided into three states (represented by the pink rectangle, yellow rectangle and green rectangle). Figure 15a–d (pink rectangle) shows that the flame has a wide diffusion shape, and it is speculated that this stage is mainly the burning of the photocurable resin. In Figure 15e–h (yellow rectangle), the flame is uniformly columnar and sprays vertically upward. At this stage, the resin burns to generate a certain amount of heat, and when the temperature reaches the decomposition temperature of AP, it will cause AP to decompose; at the same time, Al in the grain in contact with AP promotes the decomposition of AP. When Al burns, it diffuses outward together with the gas products produced by AP decomposition, and then a uniform flame with a one-way jet is produced. In Figure 15i–l (green rectangle), the flame appears to “jump”, indicating that unstable combustion occurs at this stage. The main reason for this phenomenon is that in the later stage of the grain combustion, part of the solid residue produced by the resin combustion hinders the contact between AP and Al and the path of flame stable combustion, so an irregularly shaped flame appears.

Based on the principle of target-breaking and target-passing, at 25 °C, the energetic strip with a length of 30 mm and a wide of 3 mm was tested by using the testing system shown in Figure 16, and the average value was obtained after five measurements. The burning rate test system is mainly composed of an electric detonator, a target line, a digital oscilloscope and a data processing system. According to the length of the strip and the burning time recorded by the digital oscilloscope, the linear burning rate of the energetic strip can be calculated by the formula (1). The burning rate is measured in accordance with standard GJB 772A-97, and the average burning rate was obtained after five measurements. The test results are shown in Table 6.
(1)v=L⁄t
where *v* represents the linear burning rate of the 3D-printed energetic strip in mm·s^−1^, *L* represents the distance between two target lines in mm, and *t* represents the burning time in seconds.

According to Table 6, the average burning rate of the 3D-printed strip is 5.11 mm·s^−1^, and the standard deviation is 0.32.

## 4. Conclusions

In this paper, 3D-printed energetic materials that integrate ammonium perchlorate (AP), microscopic aluminium (Al) and photocurable resin were developed. A suitable formulation of the photocurable resin and conditions in which they can be prepared as a slurry were derived. The parameters used for the production of 3D-printed shaped pellets were obtained. Based on the 3D printing technology, the additive manufacturing of special-shaped energetic grains has been successfully carried out. The mechanical and thermal properties of the new formulation are well achieved, and the printed grains are found to have high precision and fast speed.

In addition, in this paper, light-curing 3D printing technology is applied to the molding field of energetic materials, which realizes an efficient, safe, green, convenient and automatic production mode of special-shaped energetic grains. At the same time, the whole molding process is remotely controlled by a computer with reduced human participation, which better realizes human-machine isolation and ensures the safety of staff. This opens up a new method for the further development of energetic materials, which is in line with the future development concept of “green, safety and automation”. Therefore, the development of this research will be a breakthrough for the molding method of energetic materials.

In future scientific research, we can focus on how to increase the content of solid energetic fillers while meeting the conditions of the light-curing 3D printing slurry to obtain an energetic grain with superior comprehensive performance, such as in combustion and explosion, to meet different military requirements.

## Figures and Tables

**Figure 1 micromachines-12-01509-f001:**
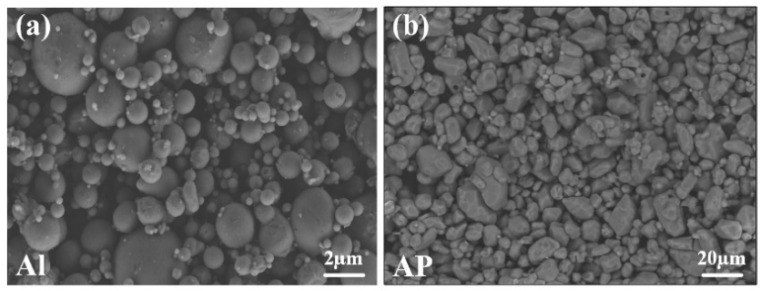
SEM of modified Al (**a**) and ultra-fine AP (**b**).

**Figure 2 micromachines-12-01509-f002:**
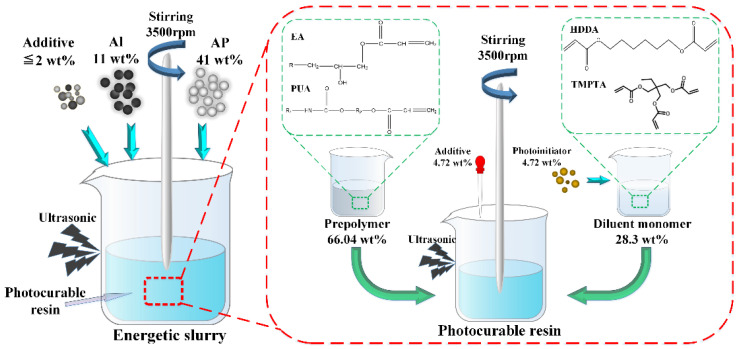
The preparation process of photocurable resin and the energetic slurry.

**Figure 3 micromachines-12-01509-f003:**
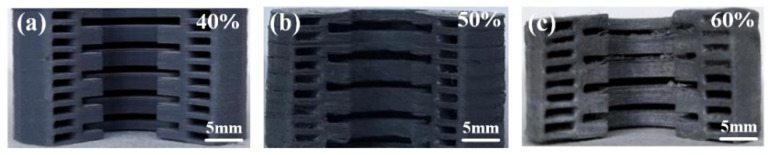
Cross-section of shaped grain samples with different solid contents: (**a**) 3D printed sample with solid content of 40%; (**b**) 3D printed sample with solid content of 50%; (**c**) 3D printed sample with solid content of 60%).

**Figure 4 micromachines-12-01509-f004:**
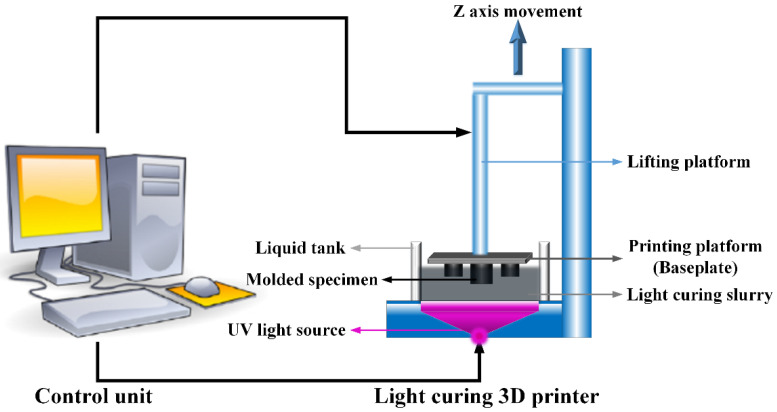
Schematic diagram of light-curing 3D printing technology.

**Figure 5 micromachines-12-01509-f005:**
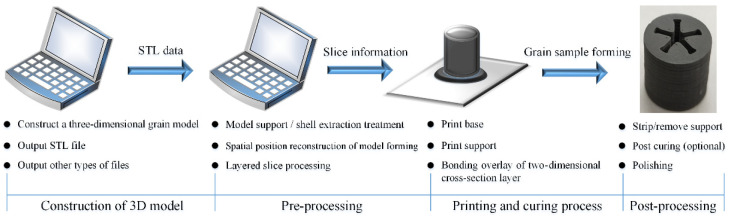
Light-curing 3D printing forming process of special-shaped energetic grains.

**Figure 6 micromachines-12-01509-f006:**
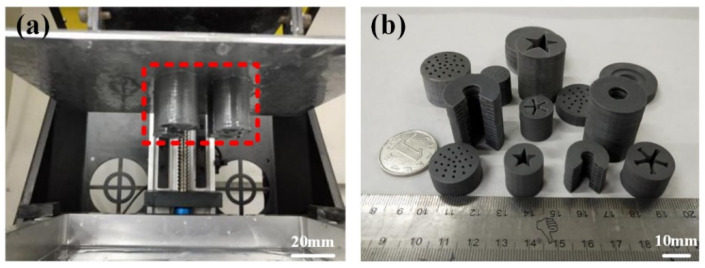
(**a**) The specific process of 3D printing. (**b**) Real samples of special-shaped energetic grains.

**Figure 7 micromachines-12-01509-f007:**
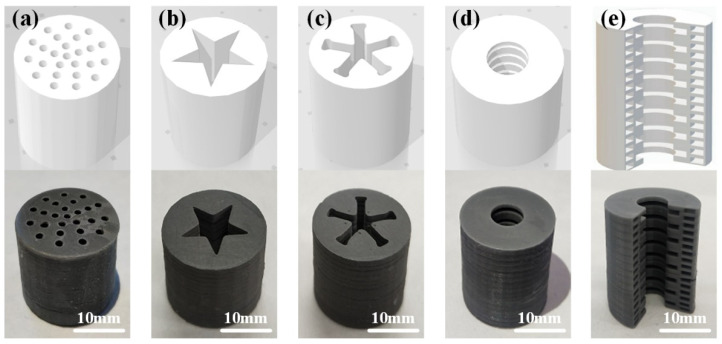
Three-dimensional CAD models of special-shaped grains and printed real samples of special-shaped grains: (**a**) Porous shape; (**b**) Star shape; (**c**) Wheel shape; (**d**) Multi-row annular hollow groove tube shape; (**e**) Cross-section of multi-row annular hollow groove tube shape.

**Figure 8 micromachines-12-01509-f008:**
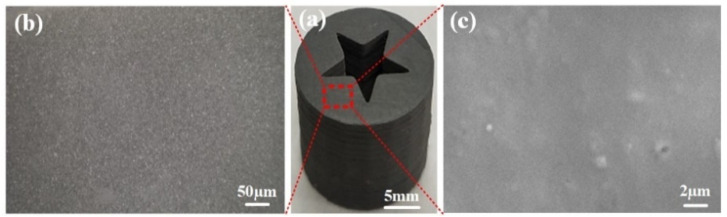
A 3D-printed energetic grain and its surface structure: (**a**) Printed grain; (**b**) Surface structure observed by optical microscope; (**c**) Surface structure observed by SEM.

**Figure 9 micromachines-12-01509-f009:**
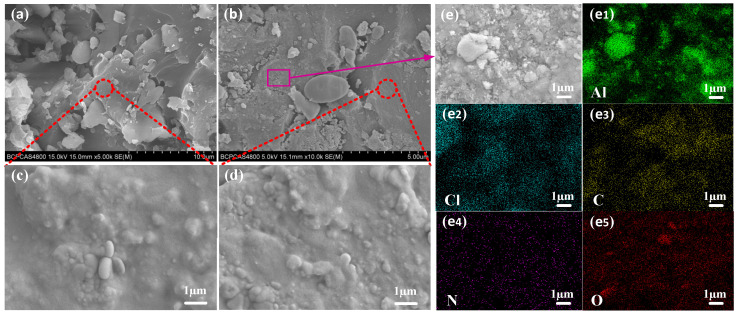
SEM images of the cross-section of the 3D-printed grain: (**a**,**b**) the micro-topography of the cross-section of the grain; (**c**,**d**) a further detail of the micro-topography of the cross-section of the grain; (**e**) SEM images of a further detail of the cross-section of grain; (**e1**–**e5**) EDS images of Al, Cl, C, N and O elements in (**e**).

**Figure 10 micromachines-12-01509-f010:**
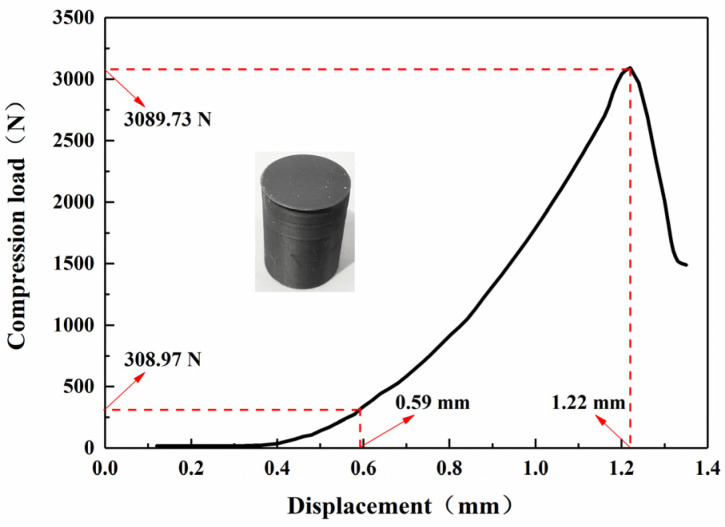
Compressive curve of a 3D-printed grain.

**Figure 11 micromachines-12-01509-f011:**
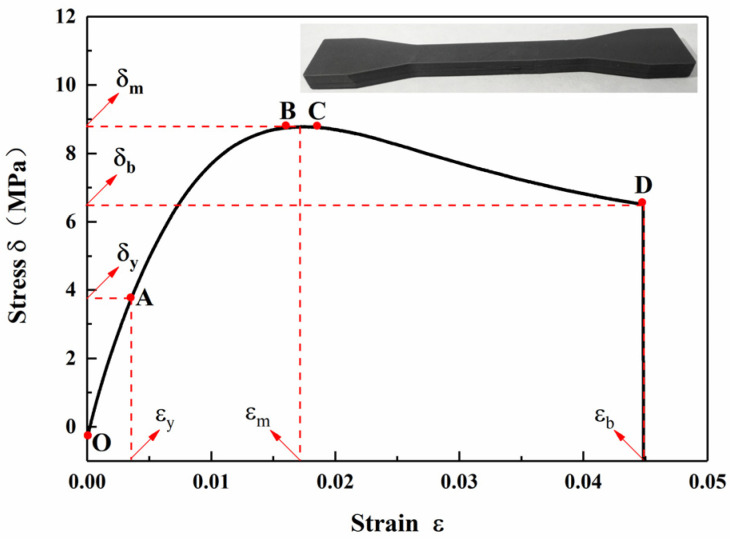
Tensile curve of a 3D-printed specimen.

**Figure 12 micromachines-12-01509-f012:**
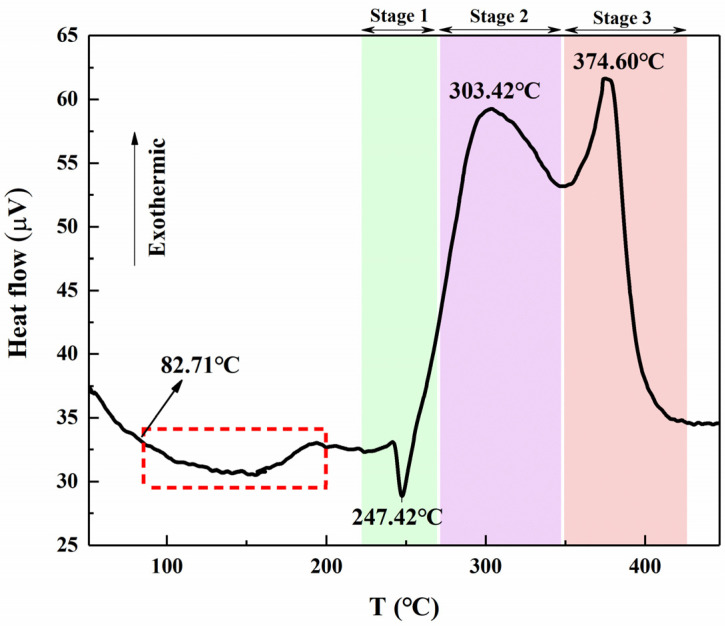
DTA curve of the 3D-printed grain.

**Figure 13 micromachines-12-01509-f013:**
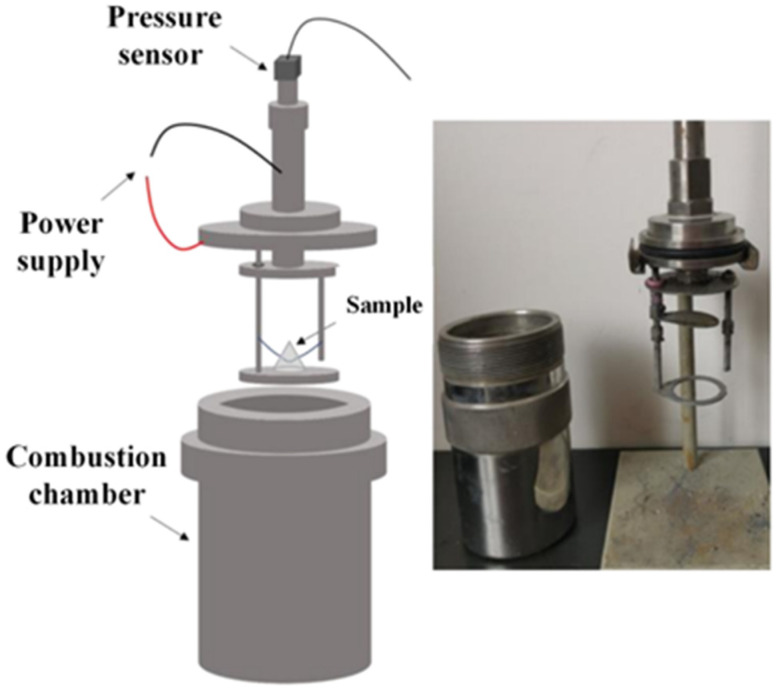
Diagram of a closed combustion chamber device.

**Figure 14 micromachines-12-01509-f014:**
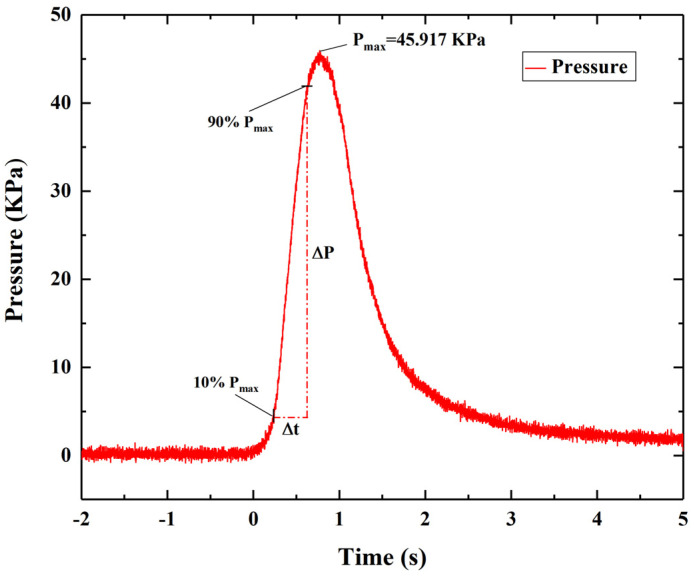
Pressure-time curve of the sample.

**Figure 15 micromachines-12-01509-f015:**
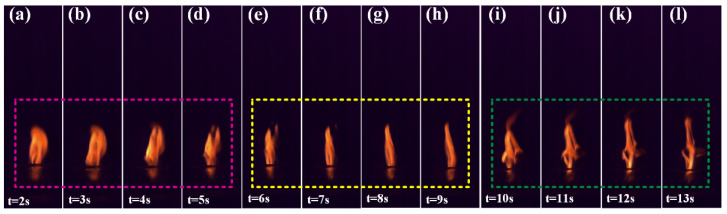
Combustion effect diagram of the 3D-printed grain at different times: (**a**) t = 2 s; (**b**) t = 3 s; (**c**) t = 4 s; (**d**) t = 5 s; (**e**) t = 6 s; (**f**) t = 7 s; (**g**) t = 8 s; (**h**) t = 9 s; (**i**) t = 10 s; (**j**) t = 11 s; (**k**) t = 12 s; (**l**) t = 13 s.

**Figure 16 micromachines-12-01509-f016:**
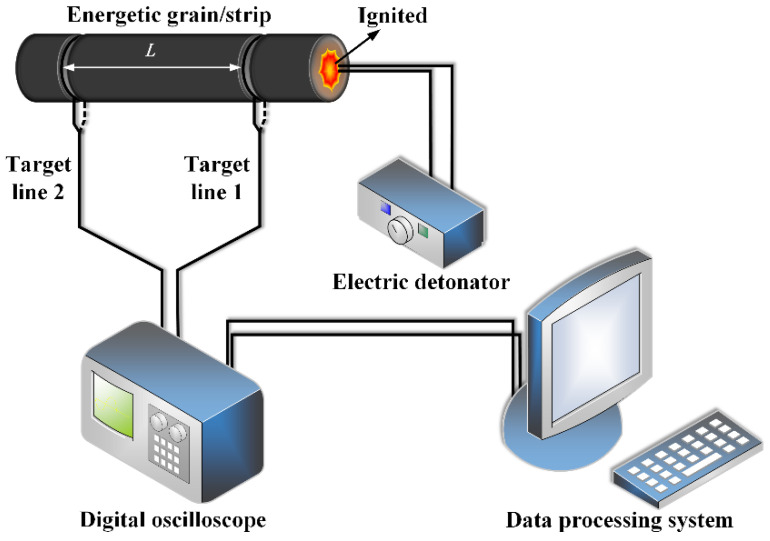
Burning rate test system of the energetic grain/strip.

**Table 1 micromachines-12-01509-t001:** The formula of photocurable resin.

Component		Fraction (wt%)
Prepolymer	EA	44.06
	PUA	21.98
Diluent monomer	HDDA	14.15
	TMPTA	14.15
TPO		4.72
Additives		0.94

**Table 2 micromachines-12-01509-t002:** Description of the energetic slurry composition.

Component	Fraction (wt%)
Photocurable resin (self-made)	48
Ultra-fine AP	41
Modified Al	11
Other additives	≤2(in addition)

**Table 3 micromachines-12-01509-t003:** Process parameters of light-curing 3D printing.

Layer	Exposure Time/ms	Up Speed mm/min	Down Speed mm/min	Lift Height/mm	Settle Time/ms
0	200,000	20	40	100	3000
1	150,000	30	50	80	2600
3	100,000	40	60	60	2200
5	50,000	50	70	40	1800
10	10,000	60	80	20	1400
20	8000	70	90	10	1000
30	7000	70	90	10	1000
40	6000	70	90	10	1000
50	6000	70	90	10	1000
60	…	…	…	…	…

**Table 4 micromachines-12-01509-t004:** Density of the 3D-printed grain.

Number	Grain Density /(g·cm^−^^3^)	Average Density /(g·cm^−^^3^)	Standard Deviation of Density /(g·cm^−^^3^)	Theoretical Density /(g·cm^−^^3^)
1	1.606	1.606	0.002	1.619
2	1.609			
3	1.604			

**Table 5 micromachines-12-01509-t005:** Data of the quasi-static mechanical properties of a 3D-printed specimen.

Compressive Strength (S_c_)/MPa	Fracture Force (F_b_)/N	Tensile Fracture Stress (σ_tb_)/MPa	Tensile Strength (σ_tM_)/MPa	Tensile Elastic Modulus (E_t_)/MPa
9.8300	259.5060	6.4877	8.7845	1027.2150

**Table 6 micromachines-12-01509-t006:** Test results of the burning rate of the 3D-printed strip.

Strip Number	Distance between Target Lines (mm)	Burning Time (s)	Burning Rate (mm·s^−^^1^)	Average Burning Rate (mm·s^−^^1^)
1	30	5.61	5.35	5.11
2	30	6.15	4.88	
3	30	5.35	5.61	
4	30	6.05	4.96	
5	30	6.34	4.73	

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
