# Peer review of "Additive Manufacturing of a Special-Shaped Energetic Grain and Its Performance"

_micromachines, 2021, doi:10.3390/mi12121509_

Round 1
Reviewer 1 Report
In the present work, Ren and coworkers report on the development of 3d printed energetic materials that integrate ammonium perchlorate, microscopic aluminium and photocurable resin. The authors have derived a suitable formulation of the photocurable resin and conditions in which they can prepare the slurry. They also report in great detail the parameters used for the production of 3D printed shaped pellets. The mechanical and thermal properties of the new formulation are well reported. Using SEM images the authors also provide an overview of the material surface.
The overall work is clear and well presented and quite solid. In the introduction, the authors provide a concise overview of the state of the art, which is very good.
The conclusion statement in a way is a repetition of the introduction. I would suggest that the authors arrange it a little bit better. For example, by providing a bigger overview on the significance of the study and in case they still intend to provide characterisation data/performance, that data/performance has to be given in comparison to other energetic systems sharing similar composition or preparation strategy.
Minor things to be corrected:
- Formatting of the reference numbering
- Do not start sentences with “And”
Author Response
Dear Referee
First of all, thank you for your approval. Thank you very much for your nice comments and valuable suggestions. All your comments are all very helpful for revising and improving our paper.
The author carefully explained the details of the revision of the manuscript, and revised and responded to all your comments. Revised portion are marked in red in the paper.
Please see the attachment.
Kind regards
Authors

Reviewer 2 Report
The manuscript under review investigates Additive manufacturing of special-shaped energetic grain and its performance. Following corrections should be incorporated before acceptance:
- Citations are provided along with the text. It may be provided as superscript or in brackets.
- English need significant improvement.
- The scale should be added in Figures 3, 6 and 7
- More evidence is required to support the line "It can be seen from Figure 9a and b that the 3D printed grains have good cross-section
integrity, most of the cross-sections are smooth, and the grain structure is relatively dense, without obvious cracks, holes and other defects" - The line "In addition, it can be clearly found from Figure 9c and d that the Al and AP particles are uniformly dispersed in the photocurable resin system" needs more clarity and evidence.
- The line " It can be seen from Figure 9 e1-e5 that the distribution of Al, Cl, C, N and O elements is relatively uniform" needs justification as segregation is seen in Fig. e1-e5.
- What is the motivation behind performing compression and tensile analysis?
- The authors may add a future scope of work and targeted applications in the conclusion.
Author Response
Dear Referee
Thank you very much for your nice comments and valuable suggestions. All your comments are all very helpful for revising and improving our paper.
The author carefully explained the details of the revision of the manuscript, and revised and responded to all your comments. Revised portion are marked in red in the paper.
Please see the attachment.
Kind regards
Authors

Round 2
Reviewer 2 Report
The manuscript may be accepted.